# Application of Liposome Encapsulating *Lactobacillus curvatus* Extract in Cosmetic Emulsion Lotion

**DOI:** 10.3390/ma14247571

**Published:** 2021-12-09

**Authors:** Hyo-Tae Kim, Jiseon Lee, Yeon-Ji Jo, Mi-Jung Choi

**Affiliations:** 1Department of Food Science and Biotechnology of Animal Resources, Konkuk University, Seoul 05029, Korea; khtleo@naver.com (H.-T.K.); jmango021@konkuk.ac.kr (J.L.); 2Department of Food Processing and Distribution, Gangneung-Wonju National University, Gangwon-do 25457, Korea; joyeonji@gwnu.ac.kr

**Keywords:** encapsulation, liposome, probiotics extract, cosmetic emulsion lotion

## Abstract

Probiotic extracts have various positive attributes, such as antioxidant, tyrosinase inhibitory, and antimicrobial activity. *Lactobacillus curvatus* produces bacteriocin, which activates the lipid membrane structure and has potential as a natural preservative for cosmetic emulsions. In this study, *L. curvatus* extract was encapsulated in liposomes and formulated as an oil-in-water (O/W) emulsion. Radical scavenging activity, tyrosinase inhibition, and challenge tests were conducted to confirm the liposome activity and the activity of the applied lotion emulsion. The liposome-encapsulated extract had a relatively high absolute ζ-potential (52.53 > 35.43), indicating its stability, and 96% permeability, which indicates its potential as an active agent in lotion emulsions. Characterization of emulsions containing the liposomes also indicated a stable state. The liposome-encapsulated extract exhibited a higher radical scavenging activity than samples without the extract and non-encapsulated samples, and the functionality was preserved in the lotion emulsion. The tyrosinase inhibition activity of the lotion emulsion with the liposome-encapsulated extract was similar to that of the non-treated extract. *Candida albicans* and *Aspergillus niger* were also inhibited in the challenge test with the lotion emulsions during storage. Collectively, these findings indicate that the liposome-encapsulated extract and the lotion containing the encapsulated extract have potential applicability as natural preservatives.

## 1. Introduction

The cosmetics industry has developed and become extremely diverse [1]. One industry trend is the use of natural cosmetics for personal care [2]. Personal care cosmetic products such as creams and lotions are based on oil-in-water (O/W) emulsions formulated with natural ingredients. However, chemical preservatives, such as sodium ethylene diamine tetraacetic acid (Na EDTA), are often added because it is difficult to safely distribute the products to consumers without preservatives.

Moreover, probiotics such as *Lactobacillus* and *Leuconostoc* have taken center-stage as cosmetic ingredients [3]. These probiotics have beneficial effects on the skin, not only through oral consumption but also through topical application. Oral consumption of probiotics can improve the overall metabolite content of the human body by inhibiting harmful gut microflora by changing the gut microbiome, which enhances the immune system [4]. Similarly, inhibiting harmful microflora by the topical application of probiotics and their extracts changes the microbiome of the skin. Probiotics and their extracts reduce skin pH through lactic acid production and the generation of an amino acid mantle, thereby facilitating moisture preservation [5]. Useful metabolites produced by probiotics are found in extracts and have various functions, such as antioxidant and tyrosinase inhibitory activity, which induce skin whitening effects [6]. Therefore, these extracts can potentially be exploited as multifunctional natural preservatives for use as cosmetic ingredients. However, these benefits of probiotics make it reasonable to use probiotics as cosmetic ingredients; however, the use of natural preservatives is still insufficient.

*Lactobacillus curvatus* produces a bacteriocin called curvacin A, which is composed of 59 amino acids and serves diverse purposes in food hygiene [7]. Studies have investigated the use of curvacin A as a starter culture and natural preservative for fermented sausage [8]. However, curvacin A requires structural activation by entrapment in lipid bilayers to induce functional antimicrobial effects [9].

Liposomes are multi-layered vesicles comprising phospholipids that can be used to encapsulate substances to protect them from hazardous environments during production processes, such as heat- and pressure-induced hazards, through homogenization [10]. Therefore, in this study, an extract of *L. curvatus* was encapsulated in liposomes to entrap curvacin A and other functional substances in the lipid membrane and inner vesicle.

Previously, bacteriocins such as nisin have been encapsulated in liposomes to achieve enhanced stability [11], which could potentially contribute to maintaining their function during the preparation of emulsions. Therefore, in this study, *L. curvatus* extract was encapsulated in phosphatidylcholine liposomes, applied to O/W lotion emulsions, and investigated as a potential multifunctional natural preservative.

## 2. Materials and Methods

### 2.1. Materials

*L. curvatus* MGB0015 was provided by the World Institute of Kimchi Microorganisms and the Gene Bank. Lipoid S75 (Lipoid, Ludwigshafen, Switzerland) and water-soluble phosphatidylcholine were used to process the liposomes. Medium-chain triglyceride (MCT) oil (Cosnet, Korea) was used as the oil phase in the emulsions. Tween 80 (Samchun, Gyeonggi-do, Korea) and oil-soluble soybean lecithin (Samchun, Korea) were used as emulsifiers in the cosmetic O/W emulsion.

### 2.2. Extraction of L. curvatus

*L. curvatus* preserved at −100 °C was reactivated in 10 mL of de Man, Rogosa, and Sharpe agar (MRS) broth medium (Neogen, MI, USA) at 30 °C for 24 h. The culture medium was further incubated with 125 mL MRS broth in a flask at 30 °C for 24 h [12] and then centrifuged at 4000 rpm for 25 min at 4 °C. The probiotics were washed twice with 30 mL of 0.85% saline water to obtain *L. curvatus* pellets, which were sonicated using an ultrasonicator (Model HD2200, Bandelin Electronic GmbH & Co. KG, Berlin, Germany). Ultrasonication was performed at 100 W for five 3-min cycles with a 2-min interval after adding 15 mL of 0.85% saline. The sonicated probiotic suspension was centrifuged at 10,000 rpm for 15 min at 4 °C. The supernatant was collected and used for subsequent experiments and is denoted as *L. curvatus* extract [13].

### 2.3. Fabrication of L. curvatus Extract Encapsulated Liposome and Characterization

#### 2.3.1. Liposome Preparation

Lipoid S75 was dissolved in the extract at 1% (*w/v*) and stirred at 700 rpm for 30 min. Lipoid S75 was also dissolved in distilled water as a control. The phosphatidylcholine solutions were homogenized at 10,000 rpm for 3 min using a high-speed homogenizer (T25 digital Ultra-Turrax^®^ high-speed mixer, KA, Staufen, Germany). The homogenized solution was sonicated using an ultrasonicator for 3 min at 60 W. 

#### 2.3.2. Zeta-Potential and Particle Size

The zeta-potential (ζ) and particle size were measured using a Zetasizer (Zetasizer Nano ZS 90, Malvern Instruments, Worcestershire, UK) to characterize the liposomes. The sample was diluted with distilled water in 200 times.

#### 2.3.3. Encapsulation Efficiency of Liposomes

The extract of *L. curvatus* was stained with 50 ppm propidium iodide (Sigma-Aldrich, St. Louis, MO, USA) and then encapsulated into the liposomes. The liposomes were centrifuged at 6500 rpm for 45 min at 4 °C. Thereafter, the supernatant was collected, and the absorbance was measured at a wavelength of 490 nm using an ultraviolet (UV) spectrophotometer (Thermo Fisher, Waltham, MA, USA). The encapsulation efficiency was calculated using Equation (1). *L*_o_ is the absorbance of the liposomes before centrifugation, and *L*_c_ is the absorbance of the liposomes after centrifugation [14].
Encapsulation efficiency (%) = 100 × *L*_c_/*L*_o._(1)

#### 2.3.4. Permeability of Liposomes

The *L. curvatus* extract was stained with 50 ppm of propidium iodide and encapsulated into liposomes mixed with 2% (*w*/*v*) Triton X-100 (Samchun, Gyeonggi-do, Korea) in a 1:1 ratio. The absorbance of the mixed suspension was then measured at a wavelength of 490 nm using a UV spectrophotometer, and the final permeability was calculated using Equation (2). *L*_o_ is the absorbance of the liposomes before treatment, and *L*_s_ is the absorbance of the liposomes after treatment with Triton X-100 [14].
Permeability (%) = 100 × *L*_s_/*L*_o._(2)

### 2.4. Preparation of Cosmetic O/W Emulsion

The water and oil phases were prepared using the compositions listed in Table 1. The oil phase was diffused into the water phase under mixing at 700 rpm for 3 min using a propeller stirrer (Chang Shin, Incheon, Korea), and the mixture was homogenized by agitation with a T25 digital Ultra-Turrax^®^ high-speed mixer at 15,000 rpm for 5 min.

### 2.5. Characterization of Microscopic Properties and Shear Stress of Cosmetic O/W Lotion Emulsion

#### 2.5.1. Characterization of Microscopic Properties

##### Zeta–Potential Measurement

The zeta-potential was measured using a Zetasizer Nano ZS 90 to characterize the O/W emulsion.

##### Droplet Size Measurement

The droplet size of the emulsion was measured using a Master sizer 3000 E (Malvern Instruments, Worcestershire, UK) to characterize the O/W emulsion. The sample was loaded to the device slowly with the certain non-sphere optical range (7~8%).

##### Optical Microscopy

Emulsion droplets were examined using optical microscopy with a CX 31RTSF microscope and a charge-coupled device camera (3.0 M, both from Olympus Optical Co. Ltd., Tokyo, Japan). Samples were observed at ×1000 magnification.

#### 2.5.2. Characterization of Shear Stress of Cosmetic O/W Emulsion

The shear stress of the emulsion was measured using a rheometer (MCR 302, Anton Paar, Graz, Austria). The diameter of the probe was 24.982 mm, and the probe was a rough surface plate probe. The shear stress was increased from 1 to 100 s^−1^ and measured three times at 25 °C.

### 2.6. Functional Test of Cosmetic O/W Emulsion

#### 2.6.1. Radical Scavenging Activity

##### DPPH

2,2-Diphenyl-1-picrylhydrazyl (DPPH, Sigma-Aldrich, St. Louis, MO, USA) was dissolved in ethanol to prepare a 0.02 mM DPPH solution; 2 mL was vortexed with 1 mL of the sample, followed by incubation for 30 min at 25 °C. A control was prepared by replacing the sample with distilled water, and a sample blank was prepared by replacing the DPPH solution with 2 mL of ethanol. The absorbance of the prepared samples was measured using a UV spectrophotometer (Thermo Fisher, Waltham, MA, USA) at a wavelength of 517 nm, and the values were calculated using Equation (3) [15]. *A*_sample_ is the absorbance of the sample mixed with the DPPH solution, and *A*_sample blank_ is the absorbance of the sample blank. BHA (butylated hydroxyanisole; Daejung, Gyeonggi-do, Korea) was used as a positive control.
Radical scavenging activity (%) = [1 − (*A*_sample_ − *A*_sample blank_/*A*_control_)] × 100.(3)

##### ABTS

2,2′-Azino-bis-(3-ethylbenzothiazoline-6-sulfonic acid) (Sigma-Aldrich, St. Louis, MO, USA) solution was dissolved in 14 mM and mixed with 5 mM potassium persulfate (Duksan, Gyeonggi-do, Korea) at a ratio of 1:1. The ABTS solution was placed in a dark room for 16 h and then diluted to obtain a solution with a UV absorbance of 0.7 at 734 nm. Thereafter, 150 µL of each sample and 150 µL of diluted ABTS solution were vortexed and incubated for 15 min at 37 °C. The absorbance of the control and the reacted solutions was measured at 734 nm and the radical scavenging was calculated using Equation (4). The control was prepared by replacing the sample with distilled water. The sample blank was prepared by using distilled water to replace the sample mixed with ABTS solution [16]. *A*_sample_ is the measured absorbance of the sample mixed with the ABTS solution, and *A*_sample blank_ is the measured absorbance of the sample blank. BHA was used as the positive control.
Radical scavenging activity (%) = [1 − (*A*_sample_ − *A*_sample blank_/*A*_control_)] × 100.(4)

#### 2.6.2. In Vitro Tyrosinase Inhibition Test

The phosphate buffer (0.1 M) was adjusted to pH 6.8, and 220 µL was mixed with 20 µL of each sample and 20 µL of in vitro mushroom tyrosinase solution (1750 U/mL, Sigma-Aldrich, Saint Louis, MO, USA). The tyrosine solution was diluted to 1.5 mM and mixed with the sample solution, followed by incubation for 15 min at 37 °C. The absorbance of the solution was measured at a UV wavelength of 475 nm, and the tyrosinase inhibition was calculated using Equation (5) [6]. *A*_sample_ is the absorbance of the tyrosinase-added sample mixture, and *C*_sample_ is the sample mixture with phosphate buffer, which replaced tyrosinase. *A*_blank_ is the absorbance of the sample mixture with phosphate buffer, which replaced tyrosine, and *C*_blank_ means *A*_blank_ with phosphate buffer, which replaced tyrosinase.
Tyrosinase inhibition (%) = 1 − [(*A*_sample_ − *C*_sample_)/(*A*_blank_/*C*_control_)] × 100.(5)

#### 2.6.3. Challenge Test

*Aspergillus niger* (KCTC 6592) and *Candida albicans* (KCTC 7122) were selected as contaminants and were subcultured on malt agar (MEA, Difco, NJ, USA) at 25 °C. The cell count of *A. niger* was adjusted to 4.60 × 10^6^ cfu/mL and that of *C. albicans* was adjusted to 2.00 × 10^5^ cfu/mL. *Escherichia coli* (KCTC 2441) and *Staphylococcus haemolyticus* (KCTC 6592) were also selected as contaminants and were subcultured in Luria-Bertani broth (Difco, NJ, USA) at 37 °C. The *E. coli* count was adjusted to 1.07 × 10^6^ cfu/mL, and that of *S. haemolyticus* was adjusted to 3.06 × 10^5^ cfu/mL. A 15 mL aliquot of the lotion emulsion was incubated with 100 µL of each contaminating microorganism. *A. niger* and *C. albicans* were incubated at 25 °C, whereas *E. coli* and *S. haemolyticus* were incubated at 37 °C. All contaminants were incubated for varying storage times (0, 1, 2, 3, and 6 d); the colonies were counted on each day [17]. *A. niger* and *C. albicans* were measured using MEA agar. *E. coli* was measured in violet-red bile agar (VRBA, Difco, NJ, USA), and *S. haemolyticus* was measured on mannitol salt agar (MSA, Difco, NJ, USA). The biological resources used in this study were obtained from KCTC.

### 2.7. Statistical Analysis

All experiments were performed in triplicate, and the reported values are the average of the replicate experiments. Statistical analysis was performed using SPSS Statistics version 24.0, for Windows (Statistical Package for the Social Science, Ver. 24.0 IBM., Chicago, IL, USA). One-way analysis of variance (ANOVA) and Duncan’s multiple range test were performed at the 95% confidence level (*p* < 0.05) to confirm significant differences among treatments.

## 3. Results and Discussion

### 3.1. Characterization of L. curvatus Extract Encapsulated Liposome

The prepared liposomes were characterized by measuring their particle size and zeta-potential. The encapsulation efficiency and permeability of the encapsulated liposomal probiotic substances were measured. The results of the liposome characterization are shown in Table 2. The ζ-potential of BL and EL had absolute values of over 30 mV, indicating a stable and repulsive state. The absolute ζ-potential of EL was significantly higher than that of the other samples (*p* < 0.05).

Encapsulated bacteriocins have cationic characteristics that cause them to interact strongly with negative charges [18]. Similarly, in another study, the absolute ζ-potential of the antimicrobial nisin encapsulated in phosphatidylcholine liposomes was found to be approximately 10 mV [19]. It seems that an additional repulsive effect changed the ζ-potential at the surface of the liposomes. Phosphatidylcholine is zwitterionic with a negative net charge [20]. Therefore, the negatively charged surfaces repel each other while interacting favorably with the cationic membrane-entrapped substances. Although the cationic charge of bacteriocin can partially reduce the repulsive interaction between the negative charges, the ratio of phosphatidylcholine on the liposome surface and partial repulsion between the liposomes and bacteriocin entrapped within them had a greater effect on the ζ-potential. The mean particle size of EL was larger than that of BL, suggesting that the liposome size was enlarged by encapsulating the extract of *L. curvatus*. Other studies of liposome-encapsulated bacteriocins showed similar average particle sizes of approximately 130–284 nm [21]. The particle size distribution is shown in Figure 1. EL had an overall larger particle size and wider graph compared with BL, and the intersection with the curve for E indicates that EL had a similar size distribution compared to that of E. These findings suggest that liposomes successfully and stably encapsulated the extract of *L. curvatus*. Although the particle size was enlarged by encapsulation of the extract, the absolute ζ-potential was higher and stable. Therefore, encapsulation of the liposomes proceeded stably.

The encapsulation efficiency of EL was lower than that of BL, and the liposome permeability of EL was larger than that of BL. It seems that the liposomes contained the encapsulated extract of probiotics, and the peptide bacteriocin was entrapped in the liposomal membrane. This process created a flexible crack, causing the encapsulated extract to move out of the internal liposome structure. This formulation showed an encapsulation efficiency of 46%, which is higher than that of other bacteriocin liposomes reported in previous studies [22,23]. The higher permeability of EL compared to BL is similar to the tendency observed for the liposomes.

### 3.2. Characterization of Liposome-Containing O/W Emulsion

#### 3.2.1. Characterization of Microscopic Properties

The prepared O/W emulsion was characterized by measuring the droplet size and ζ-potential, as shown in Table 3. The droplets of BL-L and EL-L, which contained liposomes, were larger than those of the O/W emulsions without liposomes. Similarly, the size distribution of the emulsion particles with liposomes was broader than that of the emulsion without liposomes, as shown in Figure 2.

The droplets of the emulsions with liposomes appeared to have been enlarged through partial flocculation of the liposome around the oil droplets [24].

In another study on an O/W emulsion suspended with phosphatidylcholine liposomes, a similar volume density peak and an average particle droplet size of approximately 8 µm were reported [25]. Moreover, a classical O/W emulsion made with caprylic acid (MCT oil) was compared with the corresponding Pickering emulsion, where the droplet size of the former was approximately 3 µm [26]. Moreover, the ζ-potentials of E-L and EL-L, which contain extracts of *L. curvatus*, were significantly lower than those of the emulsions without the extract (*p* < 0.05). Repulsion of the oil droplets was interrupted by proteins from the probiotic extract, such as enzymes that induced bridging flocculation and affected the ζ-potential. Similarly, EL-L had a lower absolute ζ-potential than E-L because of bridging flocculation at the surface of the oil droplets, induced by residual peptide and probiotic extract proteins that were unencapsulated during liposome fabrication [27]. Although there were differences among all the lotion samples, all the ζ-potential values were high because the polymers induced steric stabilization, as reported in a previous study [28]. However, all emulsions were formed stably with sufficient repulsion, as shown by the microscopy results in Figure 3.

For all formulations, the O/W emulsion droplets showed no morphological disruptions. These findings indicate that although there were relative differences in the droplet size and ζ-potential among formulations, stable O/W emulsions were still formed. Tween 80 (polysorbate 80) is a non-ionic surfactant that induces enhanced steric stabilization regardless of various repulsive or associative charges [29]. In addition, carrageenan, used as a thickening agent, enhances the emulsion stability according to Stoke’s law [30]. These effects were the main contributors to the stability of the prepared emulsions, regardless of whether they contained liposomes or extracts of *L. curvatus*.

#### 3.2.2. Characterization of Shear Stress

The shear stress values of the emulsions are shown in Figure 4. Generally, emulsions are pseudo-plastic fluids, and all emulsion samples show decreased viscosity when the shear rate is increased due to continuous destruction of the emulsion. Similarly, other studies on the rheology of O/W emulsions have shown pseudo-plastic characteristics [31]. In this study, the shear stress of the emulsion with liposomes was higher than that of the emulsions without liposomes under all conditions in the presence of the probiotic extract. For instance, the shear stress of BL-L was higher than that of B-L, and similarly, the shear stress of EL-L was higher than that of E-L.

The droplet size followed a similar trend and was affected by partial flocculation of the liposomes with oil droplets. A previous study showed that the increase in the viscosity is dependent on the phospholipid content of the O/W emulsion, which may account for the observed results [32]. In the initial stage when the shear rate increased, the shear stress of E-L and EL-L was higher than that of the emulsion without the probiotic extract. The proteins of the probiotic extract and carrageenan, which is a polysaccharide, appeared to act as a colloidal protein/polysaccharide coacervate, which affects the viscosity of the O/W emulsion [33]. Another study showed that an O/W emulsion coated with ovalbumin and gum arabic exhibited coacervation at a low pH of approximately 4, inducing higher viscosity than that observed under other conditions [34].

Neutralizing the charge of proteins and polymers induces coacervation [35]. The advanced partial bridging flocculation between protein/carrageenan and oil droplets induced by increasing the shear rate reduced the repulsive charge between the particles. Consequently, the adsorbed surfaces became neutrally charged, and non-ionic Tween 80 (polysorbate 80) was bound to the neutral surface. Neutral and turbid environments cause coacervation and produce sticky colloids. Therefore, the shear stress of the emulsion containing the extract of *L. curvatus* was higher than that without the extract.

### 3.3. Functional Tests of Liposome-Containing O/W Emulsion

Various beneficial attributes, such as the antioxidant, antimicrobial, and whitening activities, were analyzed to confirm the effect of the liposome-encapsulated probiotic extract in the O/W emulsions. For comparison, the extract of *L. curvatus* and the extract encapsulated in the liposomes was diluted to 63% to adjust the concentration of the emulsion components, as shown in Table 1.

#### 3.3.1. Radical Scavenging Activity

The radical scavenging activity was measured, and the results are listed in Table 4. Both the DPPH and ABTS assays showed that system E produced higher radical scavenging activity than D-E, diluted to 63% at the same concentration as that of E. Similarly, treatment EL showed higher radical scavenging activity than D-EL. This shows that the radical scavenging activity differed according to the concentration of the *L. curvatus* extract.

The radical scavenging activity of D-EL was significantly higher than that of D-E, and the radical scavenging activity of EL was significantly higher than that of E (*p* < 0.05). The superoxide dismutase present in the probiotic extract was protected from the hazardous environment and showed enzymatic activity that increased the radical scavenging effects [36]. Similarly, a glutathione-containing probiotic extract was encapsulated in liposomes, which plausibly enhanced the radical scavenging activity [37]. Herein, liposome encapsulation produced a comparable effect, similar to the results reported in the study of the antioxidant activity of liposome-encapsulated curcumin [38].

Comparison of the data for the O/W emulsions showed that E-L and EL-L exhibited higher radical scavenging activity than B-L and EL-L, which showed higher activity than E-L in the ABTS assay. Furthermore, EL-L showed higher radical scavenging activity than D-EL with the same concentration of *L. curvatus* extract in the formulated emulsions in both assays (DPPH and ABTS). Moreover, the difference was significant in the ABTS assay (*p* < 0.05). These findings indicate that the protective effect of liposomal encapsulation was maintained, which diminished the effect of external pressure and heat during preparation of the emulsion. Although encapsulation showed a protective effect, the liposomes prepared with the probiotic extract had relatively high permeability and radical scavenging activity.

#### 3.3.2. In Vitro Tyrosinase Inhibition

*Lactobacillus* species exhibit tyrosinase inhibitory activity [39]; therefore, *L. curvatus* extract was analyzed in this study. The results of the in vitro tyrosinase inhibition tests are shown in Figure 5.

Treatment E afforded significantly higher tyrosinase inhibition than that of D-E (*p* < 0.05). This result suggests that the inhibition of tyrosinase changed depending on the concentration of the *L. curvatus* extract. Similarly, EL showed significantly higher tyrosinase inhibition activity than D-EL (*p* < 0.05). Moreover, EL, similar to the liposome-encapsulated extract, showed lower tyrosinase inhibition than E alone.

Generally, substances with tyrosinase inhibitory and whitening effects in wide commercial use are nucleic acids, such as adenosine [40]. The encapsulation efficiency of the liposomes was determined by treatment with propidium iodide to stain the nucleic acid for evaluation. The extract of the probiotics was not fully encapsulated in the liposomes, leading to loss. Therefore, EL showed lower tyrosinase inhibitory activity than E.

The tyrosinase inhibitory activity of psoralen and resveratrol entrapped in the liposomes showed a similar tendency depending on the entrapment efficiency [41]. However, comparison of the emulsion samples showed that the tyrosinase inhibitory activity changed in the following order: EL-L > EL > BL. Furthermore, the differences were significant among the treatments (*p* < 0.05). These results show that encapsulation with liposomes protected the probiotic extract from the effects of high rotating speed, which generates heat and pressure that damages the tyrosinase inhibitors. In addition, the high permeability of the liposomes affected EL-L, for which the tyrosinase inhibitory activity was similar to that of the original *L. curvatus* extract.

#### 3.3.3. Challenge Test

The challenge test results for bacteria, fungi, and yeast are shown in Figure 6 and Figure 7.

Over two days, the *E. coli* colonies increased in the lotions formulated without the probiotic, and were maintained for six days, whereas in the lotions containing the extract, the number of colonies formed over two days was less. In particular, EL-L, which contained the liposome-encapsulated extract, contained fewer colonies than E-L. However, both lotions containing the extract showed an increase in the number of colonies after two days, with or without liposome encapsulation. Encapsulation of curvacin A in the liposome membrane appeared to slightly affect *E. coli*, but only for two days. A previous study showed that curvacin A has low antimicrobial effects on *E. coli* [42]. In E-L, the colonies were maintained for up to two days, possibly because the lactic acid was over-diluted during the emulsion preparation process.

For *S. haemolyticus*, all lotions showed an increase in the number of colonies for up to six days of storage, but EL-L contained fewer colonies than the other lotions. Structurally activated curvacin A appeared to disturb the growth of *S. haemolyticus*, with remarkable antimicrobial effects, but did not completely destroy the colonies. A previous study of combined antimicrobial substances, including curvacin A, showed a decrease in the Gram-positive *Staphylococcus,* but the colonies were not completely destroyed [43]. In contrast, the emulsions showed more marked antimicrobial effects against fungi and yeast than the bacteria. *A. niger* and *C. albicans* are representative contaminants found in cosmetics and have been set up as the criteria for the standards of the European Pharmacopeia Commission [44]. In the case of the emulsion with liposome-encapsulated probiotic extract (EL-L), the *A. niger* colonies were completely destroyed. It seems that membrane-structured curvacin A exerts superior antimicrobial effects against fungi [45]. A previous study on the antifungal activity of various bacteriocins and curvacin A also showed antifungal effects on *A. niger* [46]. Similarly, EL-L completely inhibited the growth of *C. albicans* colonies, whereas the other emulsions did not.

## 4. Conclusions

*L. curvatus* was extracted to obtain probiotic extracts with various functions, such as antioxidant and tyrosinase inhibition, and curvacin A. The extract was encapsulated with liposomes to protect it from the harsh environment during preparation while maintaining its activity and activating curvacin A. The liposome was formulated into an O/W emulsion that maintained its activity and emulsion characteristics. The particle size and ζ-potential of the liposome were stable.

Analysis of the encapsulation efficiency and permeability indicated that the functionality of the encapsulated extract was preserved during preparation of the liposomes and emulsions. The emulsions with or without liposomes containing the probiotic extract were stably processed. The radical scavenging activity assay showed that the extract of probiotics had a concentration-dependent effect, while liposomes with the extract and the emulsion containing the encapsulated extract maintained their function. The tyrosinase inhibition followed a trend similar to that of the radical scavenging activity. Furthermore, the challenge antimicrobial test showed that the encapsulated extract did not have marked antibacterial effects, but exhibited potent activity against fungi and yeast. Therefore, encapsulating the *L. curvatus* extract within liposomes is a prospectively useful strategy for retaining its multifunctional natural preservative effect in O/W emulsions, but further studies on the potential antibacterial effects are required.

## Figures and Tables

**Figure 1 materials-14-07571-f001:**
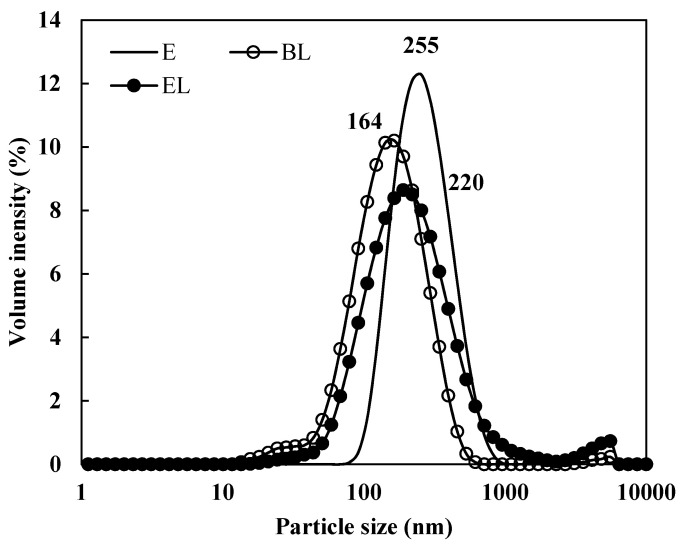
Volume intensity dispersion of liposome particle size. E: extract of *L. curvatus*; BL: blank liposome; EL: liposome-encapsulated extract of *L. curvatus*.

**Figure 2 materials-14-07571-f002:**
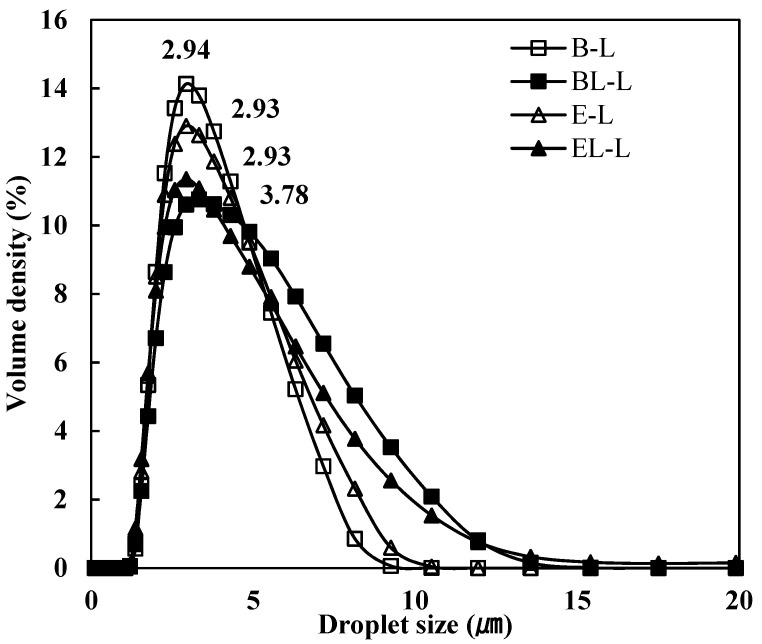
Droplet size distribution of O/W emulsions. B-L, blank lotion; BL-L, lotion with blank liposome; E-L, lotion with *L. curvatus* extract; EL-L, lotion with *L. curvatus* extract encapsulated with liposomes.

**Figure 3 materials-14-07571-f003:**
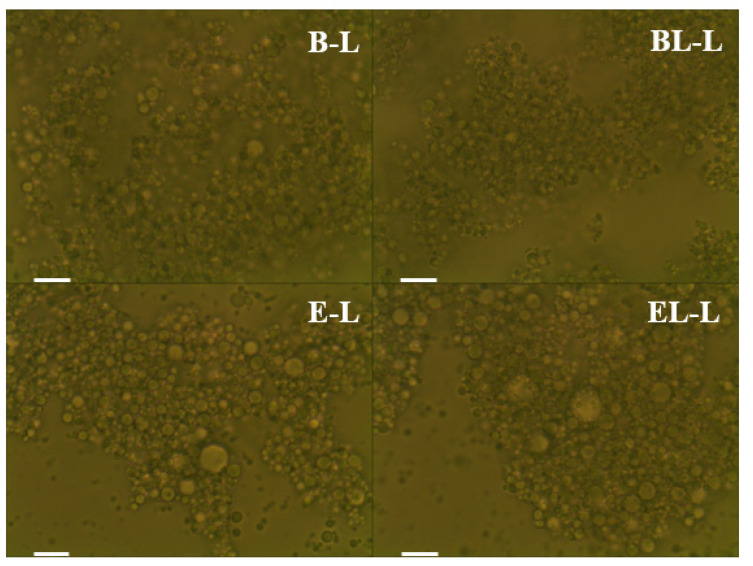
Optical microscopy image of O/W emulsion lotions. B-L, blank lotion; BL-L, lotion with blank liposome; E-L, lotion with *L. curvatus* extract; EL-L, lotion with *L. curvatus* extract encapsulated with liposomes. The scale bar on the left side of each picture is 40 μm; ×1000 magnification.

**Figure 4 materials-14-07571-f004:**
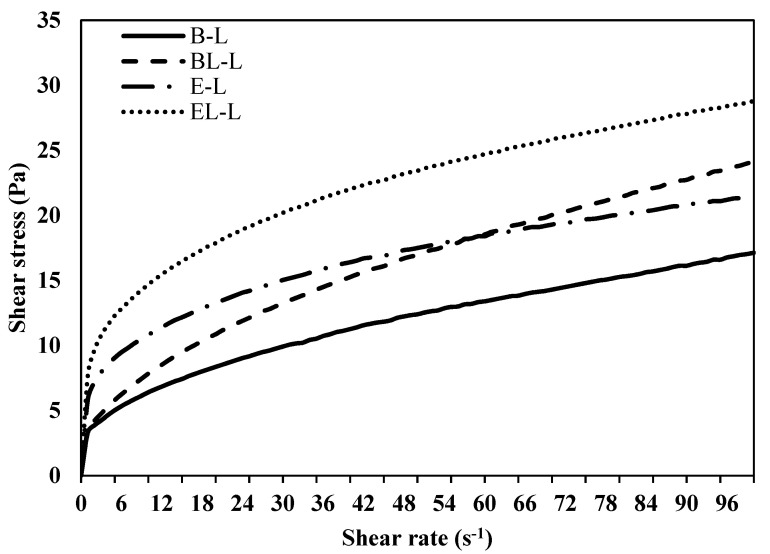
Shear stress of O/W emulsions with increasing shear rate. B-L, blank lotion; BL-L, lotion with blank liposome; E-L, lotion with *L. curvatus* extract; EL-L, lotion with *L. curvatus* extract encapsulated with liposomes.

**Figure 5 materials-14-07571-f005:**
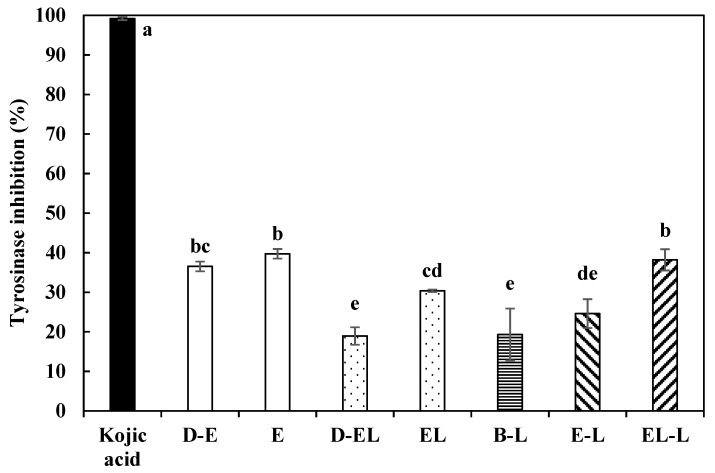
Tyrosinase inhibition of *Lactobacillus curvatus* extract in various forms. D-E: extract of *L. curvatus* diluted to 62%; E: extract of *Lactobacillus curvatus*; D-EL: liposome-encapsulated extract of *L. curvatus* diluted to 62% EL: liposome-encapsulated extract of *L. curvatus*; B-L: blank lotion; E-L: lotion with *L. curvatus* extract; EL-L: lotion with *L. curvatus* extract encapsulated with liposomes. Kojic acid was used as the positive control. ^a–e^ Indicates a significant difference among the samples (*p* < 0.05).

**Figure 6 materials-14-07571-f006:**
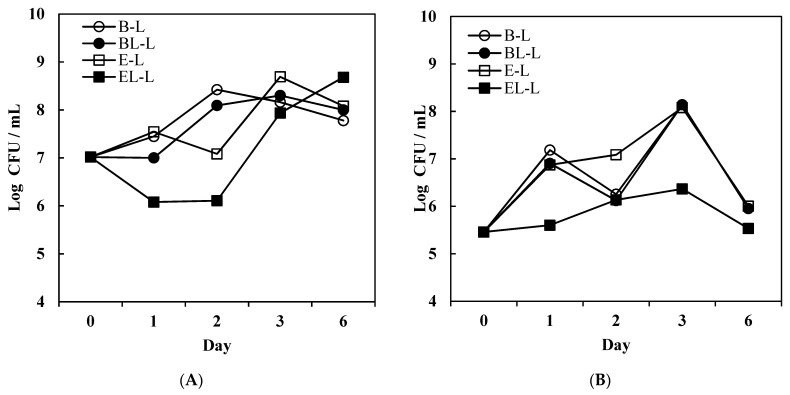
Change in number of colonies in challenge test for O/W emulsions during storage. Effects on (**A**) *E. coli* and (**B**) *S. haemolyticus.* B-L, blank lotion; BL-L, lotion with blank liposomes; E-L, lotion with *L. curvatus* extract; EL-L, lotion with *L. curvatus* extract encapsulated with liposomes.

**Figure 7 materials-14-07571-f007:**
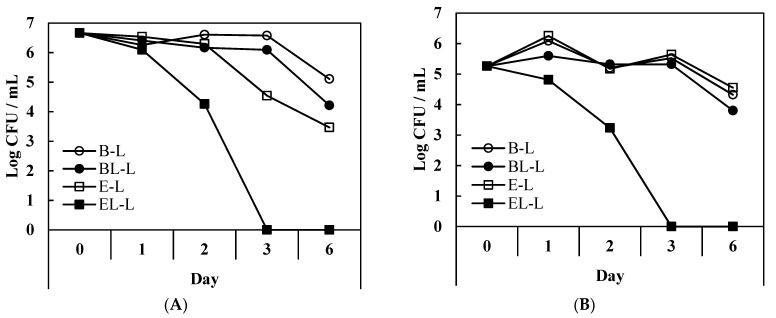
Change in number of colonies in challenge test for O/W emulsions during storage. Effects of (**A**) *A. niger* and (**B**) *C. albicans.* B-L, blank lotion; BL-L, lotion with blank liposome; E-L, lotion with *L. curvatus* extract; EL-L, lotion with *L. curvatus* extract encapsulated with liposomes.

**Table 1 materials-14-07571-t001:** Composition of cosmetic O/W emulsion.

Treatments	Composition (%)								
	Water Phase							Oil Phase	
	Water	Liposome	Extract	Liposome Coating Extract	Glycerin	Tween 80	Carrageenan	MCT Oil	Lecithin
B-L	62.0	-	-	-	7.0	0.7	0.3	28.5	1.5
BL-L	-	62.0	-	-	7.0	0.7	0.3	28.5	1.5
E-L	-	-	62.0	-	7.0	0.7	0.3	28.5	1.5
EL-L	-	-	-	62.0	7.0	0.7	0.3	28.5	1.5

B-L, blank lotion; BL-L, lotion with blank liposome; E-L, lotion with *L. curvatus* extract; EL-L, lotion with *L. curvatus* extract encapsulated with liposomes.

**Table 2 materials-14-07571-t002:** Characterization of liposomes.

Treatments	Particle Size (nm)	PdI	[-] ζ-Potential (mV)	Encapsulation Efficiency (%)	Permeability (%)
E	240.92 ± 1.95 ^c^	0.23 ± 0.01 ^a^	10.40 ± 0.16 ^c^	-	-
BL	125.60 ± 0.65 ^b^	0.26 ± 0.00 ^c^	35.43 ± 0.26 ^b^	98.09 ± 1.13	60.29 ± 0.92
EL	174.80 ± 3.37 ^a^	0.34 ± 0.02 ^b^	52.53 ± 0.52 ^a^	52.46 ± 0.63	96.71 ± 1.07

E, extract of *L. curvatus*; BL, blank liposome; EL, liposome-encapsulated extract of *L. curvatus.*
^a–c^ Means with the same letter in a column are not significantly different at *p* < 0.05.

**Table 3 materials-14-07571-t003:** Droplet size and ζ-potential of O/W emulsion.

Treatments	Droplet Size *d* _[4,3]_ (µm)	[-] ζ-Potential (mV)
B-L	3.54 ± 0.02 ^c^	58.57 ± 0.33 ^ab^
BL-L	4.37 ± 0.18 ^a^	59.33 ± 0.59 ^a^
E-L	3.69 ± 0.03 ^b^	58.47 ± 0.24 ^ab^
EL-L	4.23 ± 0.19 ^a^	58.23 ± 0.21 ^b^

B-L, blank lotion; BL-L, lotion with blank liposome; E-L, lotion with *L. curvatus* extract; EL-L, lotion with *L. curvatus* extract encapsulated with liposomes. ^a–c^ Means with the same letter in a column are not significantly different at *p* < 0.05.

**Table 4 materials-14-07571-t004:** Radical scavenging activity of *L. curvatus* extract in various forms.

Treatments	DPPH (%)	ABTS (%)
BHA	81.76 ± 1.44 ^a^	97.16 ± 0.45
D-E	5.57 ± 0.74 ^c^	46.77 ± 1.61
E	6.29 ± 0.68 ^c^	52.97 ± 1.79
D-EL	59.40 ± 3.58 ^b^	48.00 ± 3.25
EL	58.36 ± 6.42 ^a^	73.42 ± 1.28
B-L	65.20 ± 8.41 ^b^	9.30 ± 6.02
E-L	54.32 ± 10.73 ^b^	35.14 ± 15.76
EL-L	60.63 ± 7.50 ^b^	68.88 ± 8.84

DPPH, 2,2-diphenyl-1-picrylhydrazyl; ABTS, 2,2′-azino-bis(3-ethylbenzothiazoline-6-sulfonic acid); BHA: butylated hydroxy anisole; D-E: extract of *L. curvatus* diluted to 63%; E: extract of *L. curvatus*; D-EL: liposome-coated extract of *L. curvatus* diluted to 63%; EL: liposome-encapsulated extract of *L. curvatus*; B-L: blank lotion; E-L: lotion with *L. curvatus* extract; EL-L: lotion with *L. curvatus* extract encapsulated in liposomes. BHA was used as a positive control for radical scavenging activity. ^a–c^ Means with the same letter in a column are not significantly different at *p* < 0.05.

## Data Availability

Not applicable.

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
