# Peer review of "Application of Liposome Encapsulating Lactobacillus curvatus Extract in Cosmetic Emulsion Lotion"

_materials, 2021, doi:10.3390/ma14247571_

Round 1
Reviewer 1 Report
Authors proposed a paper entitled: “Application of Lactobacillus curvatus extract encapsulated liposome to cosmetic emulsion lotion”
The paper lacks of proper information and needs to be revised. Here is the list of the issues:
Line 12, “structure has potential”; add a comma among “structure” and “has potential”.
Line 17 “ higher absolute value of ζ-potential”, define this value in the abstract.
Line 68. “Lipoid S75 (Gmbh, Ludwigshafen, Switzerland”. Which is the name of the manufacturer? “Gmbh” is not a society, and Ludwigshafen is in Germany. Please, check the correct manufacturer of Lipoid S75.
Line 77. “was further incubated again” please rephrase this sentence.
Line 89. “It was also dissolved in distilled water as control” Are authors dissolving phospholipids into water? in this case, this is not possible, phospholipids are not soluble in water, but in organic solvents such as ethanol.
How do the authors prepare liposomes? No description on the process has been provided.
Line 103. “The values were then calculated using the following formula. Lo is the measurement of liposome before centrifuge, and Lc is the measurement of liposome after centrifuge”. Authors perhaps did not describe in a proper manner this part about the encapsulation efficiency. They should not just say “measurement of liposomes”, they should talk about “absorbance” measured on the supernatant. Supernatant is obtained after separation of solid pellet and aqueous part of the produced liposomes; therefore, how authors measured Lo and how Lc? Please, be more specific and clarify this section.
Table 2. I would like authors to write how they determined the standard deviations of their samples. Are these values errors or are they standard deviations? 1.95 nm is too little for a stardard deviation, having a mean value of 241 nm and a PDI of 0.23. Did you compare these standard deviations with particle size distribution diagrams shown in figure 1?
Line 223. “surficial” should be “superficial”
Line 229. “0.26 and 0.34 each, which means the peak of volume intensity was mean and mono diverse.” These values of PDI are not generally linked to monodispersed samples; generally, PDI lower than 0.2 are. What did you mean with this sentence? please, clarify.
Figure 1. I suggest adding “ticks” to diagram axes.
Figure 1. Could you please comment the small peak at large diameters, among 5000 and 10000 nm?
I suggest adding an abbreviation list, according to the guidelines of this journal, including BL-L, EL-L, O/W etc.
Table 3. Standard deviations; here there is the same problem shown in table 2.
Figure 3. Reference bars are not visible in these images.
Author Response
The paper deals with the influence of probiotics extract encapsulated liposome on the cosmetic emulsion lotion as multifunctional preservative. For this purpose, several experimental techniques were used. The research topic falls within the scope of the Materials and could be of the scientific interest. Nevertheless, the content of the manuscript evidence some lacks that must be improved. It is a well-written and comprehensible paper, however results sections need to be improved. Deeper discussion and comparison with previous works should be included in results section. Also, overall English was revised in manuscript for enhancing the quality.

Reviewer 2 Report
The authors presented an interesting study of liposome-encapsulated functional extract, prospectively relevant for contemporary cosmetic formulations. However, certain aspects of the study are presented with insufficient detail.
-The entire manuscript needs thorough language editing. Some parts are hard to follow. Also, the terminology should be uniform and precise (e.g. emulsion lotion is more correct than lotion emulsion).
-Although a reference is given, the readers would benefit from additional information on the reasons behind the need for curvacin A’s structural activation. Please elaborate this structure-function paradigm.
-line 75: please correct the section 2.2 (e.g. it says: Preserved L. curvatus preserved at -100℃ was reactivated…)
-I do not understand the title of the section 2.3. ‘Process of L. curvatus extract encapsulated liposome and the characterization’ Please rewrite.
-section 2.3.1 must be written more clearly and precisely.
-more information on the measurements within section 2.3.2 are needed. Was the conductivity controlled during ZP measurements? What dilution was applied?
-please provide uniformly throughout the manuscript information on the substances or equipment used (e.g. trade name, producer, city, state). Please consult the Journal’s propositions.
-section 2.4: Were the cosmetic emulsions prepared without heating of the water and oil phases?
-section 2.5.1.1: The Zetasizer nano ZS 90 is not suitable for the emulsions you prepared, since the droplet sizes are outside of the instrument’s optimal range. You must have applied a diluted sample, thus completely changing the structure of the O/W emulsion.
-section 2.5.1.2. Detailed conditions of the measurement need to be disclosed.
-line 223: surficial?
-lines 226-228: please rephrase the sentence to make it more clear.
-lines 237-239: something is missing – please rephrase both sentences.
-Figure 1. only the legend for E (line) and BL (empty circle) are given. Please add the explanation for black circles curve.
-line 244: ‘Encapsulation efficiency of EL was lower than BL’ – According to method description, encapsulation efficiency was only done for EL? Please explain.
-line 255: ‘The emulsion characterization was microscopically characterized to confirm the suitability of the O/W emulsion.’ – Unfortunately, microscopy is not enough to confirm emulsion stability, so please rephrase or erase this statement.
-Figure 3: the presented micrographs do not support the emulsion stability the authors implied. Please provide better proof.
Author Response

(The authors gave the same response as above.)

Round 2
Reviewer 1 Report
Line 67. “Lipoid S75 (Lipoid, Ludwigshafen, Switzerland) and water-soluble phosphatidylcholine”. I know that Lipoid produces formulation of phospholipids for the preparation of liposomes. In particular, I am using Lipoid S 45, that is characterized by 45 % phosphatidylcholine. Generally I dissolve this product in ethanol. Do you believe that it is totally soluble in water ?
Table 2. I still have doubts about this standard deviations results. 1.95 nm, 0.65 nm, 3.37 nm are too little to justify PDI of 0.23 to 0.30. May the authors refer to errors, and not to standard deviations?
Line 238. “the absolute zeta-potential was higher and stable” what does it mean that the absolute zeta-potential is stable? over time observation?
Figure 3. Reference bar is not particularly visible.
Author Response
The revision letter was attached

Reviewer 2 Report
Although the authors addressed the majority of the queries from the 1st round of review, I have some additional remarks. English language editing is still needed throughout the text, and there are a number of technical imperfections:
-line 16 (abstract): why did you write 52.53>35.43?
-section 2.5.11: again, I assume dilution was necessary for ZP measurements?
-section 2.5.2: lines 145-146: at 25 an?
-line 167: at 37° a?
-line 191: at 37° L?
-line 194: at 37° w?
-lines 290-291: I disagree that the presented micrographs are in favor of emulsion stability. Please replace them with other more suitable micrographs, or erase the Figure 3 altogether.
Author Response
The revision letter was attached

This manuscript is a resubmission of an earlier submission. The following is a list of the peer review reports and author responses from that submission.